# Dietary Supplementation of Novel Aflatoxin Oxidase CotA Alleviates Aflatoxin B_1_-Induced Oxidative Stress, Lipid Metabolism Disorder, and Apoptosis in the Liver of Japanese Quails

**DOI:** 10.3390/ani15111555

**Published:** 2025-05-26

**Authors:** Hao Lv, Zhiyong Rao, Yuting Li, Wei Zhang, Lihong Zhao, Zhixiang Wang, Yongpeng Guo

**Affiliations:** 1College of Animal Science and Technology, Henan Agricultural University, Zhengzhou 450046, China; lcxhau@163.com (H.L.); rzyhau@163.com (Z.R.); liyuting018@163.com (Y.L.); weizhang@henau.edu.cn (W.Z.); wzxhau@henau.edu.cn (Z.W.); 2College of Animal Science and Technology, China Agricultural University, Beijing 100193, China; zhaolihongcau@cau.edu.cn

**Keywords:** aflatoxin B_1_, aflatoxin oxidase CotA, liver damage, Japanese quails

## Abstract

Aflatoxin B_1_ frequently contaminates animal feed, posing a considerable risk to the poultry industry. There is an urgent necessity to develop effective strategies for mitigating the harmful effects of aflatoxin B_1_ on poultry health. This study aimed at assessing the efficacy of dietary supplementation with the novel aflatoxin oxidase CotA in mitigating the adverse effects of an aflatoxin B_1_-contaminated diet in Japanese quails. The findings indicate that aflatoxin oxidase CotA effectively detoxified dietary aflatoxin B_1_-induced hepatotoxicity and reduced liver aflatoxin B_1_ residues in Japanese quails. These results underscore the potential application of aflatoxin oxidase CotA as a promising agent for aflatoxin B_1_ degradation in the feed industry.

## 1. Introduction

Aflatoxins are a class of toxic secondary metabolites mainly produced by *Aspergillus parasiticus* and *Aspergillus flavus*. Aflatoxins were initially identified in 1961 by Van der Zijden et al. [1] as the primary etiological agent of “Turkey X disease”, which led to the deaths of more than 100,000 turkeys [2]. To date, over 20 types of aflatoxins have been discovered. Among them, aflatoxin B_1_ (AFB_1_) exhibits the highest toxicity and is classified as a Group I carcinogen by the International Agency for Research on Cancer (IARC) [3]. In the liver, aflatoxin B_1_ (AFB_1_) undergoes bioactivation through phase I metabolic enzymes, specifically cytochrome P450 (CYP450), resulting in the formation of exo-AFB_1_-8,9-epoxide (AFBO) [4]. This epoxide subsequently forms adducts with DNA, leading to cytotoxicity, mutation, and DNA damage [5]. Poultry, including chickens, ducks, turkeys, and quails, are among the animal species that are vulnerable to the toxic impacts of AFB_1_. Dietary AFB_1_ exposure can lead to serious organ damage, reduced production performance, immune suppression, intestinal microbiota disorder, increased morbidity and mortality rates, and meat quality deterioration in poultry [6]. There is an urgent necessity to explore effective strategies for preventing or alleviating the deleterious impacts of AFB_1_ on poultry [7].

The utilization of mycotoxin binders for preventing aflatoxicosis has gained prominence in poultry production. Numerous studies have demonstrated that mineral and organic adsorbents, including zeolite, bentonite, aluminosilicate, montmorillonite, and yeast cell wall, can partially or completely mitigate the deleterious effects of dietary AFB_1_ [8]. Nevertheless, these adsorbents may also sequester essential nutrients such as minerals and vitamins. Biological degradation, characterized by the microbial and enzymatic conversion of AFB_1_ into less harmful metabolites, is regarded as a promising approach due to its high specificity, efficiency, and environmental sustainability. Various enzymes, including peroxidases [9,10], laccases [11,12], N-acyl endolipases [13], and F_420_H_2_-dependent reductases [14], have been identified as capable of degrading AFB_1_. Currently, research on AFB_1_-degrading enzymes predominantly concentrates on verifying AFB_1_ degradation in vitro, with relatively few in vivo studies evaluating the efficacy of these enzymes in animal production [15]. Our previous research established that CotA laccase derived from *Bacillus licheniformis* exhibited aflatoxin oxidase activity, which could catalyze the C3-hydroxylation of AFB_1_ to non-toxic aflatoxin Q_1_ and epi-aflatoxin Q_1_ [12]. Moreover, CotA laccase significantly enhanced growth performance, intestinal health, and hepatic AFB_1_ metabolism in ducks consuming a diet naturally contaminated with AFB_1_. Dietary supplementation of CotA reduced AFB_1_-DNA adducts in the liver and decreased AFB_1_ residues in both the liver and feces [16]. These findings underscore the potential of CotA as a safe and effective aflatoxin oxidase.

Raising Japanese quails has become increasingly popular within the poultry industry, especially in developing countries, as an alternative protein source for humans. Previous research has demonstrated that the intake of diets contaminated with AFB_1_ adversely affects hepatic function in Japanese quails, leading to a loss of production performance and reduction in egg weight [17]. Additional studies have reported the immunosuppressive effects of AFB_1_ exposure in Japanese quails [18,19]. However, there is limited information available on biological detoxification strategies for quail diets contaminated with AFB_1_. Therefore, the current study was conducted to examine the protective effects of the novel aflatoxin oxidase CotA in mitigating AFB_1_-induced liver damage in Japanese quails.

## 2. Materials and Methods

### 2.1. Materials and Animal Trial Design

AFB_1_ (purity ≥ 98%) was procured from Pribolab Biological Technical Company, Qingdao, China. The aflatoxin oxidase CotA from *Bacillus licheniformis* ANSB821 was expressed in *Pichia pastoris* and subsequently freeze-dried under vacuum conditions for 24 h [12,20]. Japanese quail chicks (1 week of age, female) were purchased from Henan Aoxiang Quail Breeding Base (Henan, China). After acclimation for 1 week, a total of 225 Japanese quail chicks were randomly allocated into three groups, each comprising five replicates with 15 quail chicks per replicate. The experimental groups included: (1) a basal diet serving as the control group (CON); (2) a basal diet with 50 μg kg^−1^ of AFB_1_ (AFB_1_ group); and (3) a basal diet with 50 μg kg^−1^ of AFB_1_ and 1 U kg^−1^ of aflatoxin oxidase CotA (AFB_1_+CotA group). The AFB_1_ level in the diet and the dose of aflatoxin oxidase CotA included in this study were set according to our previous study [16]. The basal diet was formulated following the NRC (1994) guidelines for growing Japanese quails (Appendix A). The birds had free access to feed and water. Quails within each replicate were housed in cages measuring 90 × 40 × 40 cm, with the ambient temperature being maintained at 25 °C ± 2 °C throughout the duration of the 21-day feeding trial.

### 2.2. Growth Performance and Sample Collection

Upon the completion of the feeding trial, the quails were fasted for 12 h and weighed to determine average daily gain (ADG), average daily feed intake (ADFI), and feed conversion ratio (FCR). Subsequently, three birds from each replicate were selected for the collection of blood and tissue samples. Blood samples were taken from the brachial vein, clotted at room temperature for 2 h, and centrifuged at 3000 rpm for 10 min to separate the serum. The serum was then stored at −80 °C for subsequent analysis. Following blood collection, the birds were dissected, and whole liver tissues were promptly isolated and blotted with absorbent paper to eliminate surface moisture. The liver index was calculated by measuring liver weight with an electronic balance. A portion of the liver samples was fixed in 4% paraformaldehyde, while another portion was stored at −80 °C for further analysis.

### 2.3. H&E Staining

Liver tissue specimens were initially fixed in 4% paraformaldehyde, followed by dehydration in ethanol, and subsequently embedded in paraffin. The specimens were sectioned into slices of 5 µm thickness and stained with hematoxylin and eosin (H&E) for pathological examination using a digital microscopy scanner (3DHistech Ltd., Budapest, Hungary).

### 2.4. Oil Red O Staining

To assess hepatic lipid droplet deposition, the Oil Red O staining method was employed. Liver tissues were fixed in 4% paraformaldehyde, embedded in the Tissue-Tek O.C.T. compound, and sectioned into slices approximately 5 μm thick. The sections were then stained with Oil Red O and counterstained with hematoxylin, with the visualization of the red lipid droplets performed using a Nikon microscope (Nikon, Melville, NY, USA).

### 2.5. TUNEL Staining

Apoptosis in liver tissues was evaluated using TUNEL staining. Briefly, liver tissues were fixed in 4% paraformaldehyde, dehydrated, embedded in paraffin, and dewaxed post-sectioning. The sections were permeabilized with proteinase K, followed by labeling with BrightRed Labeling Mix at 37 °C for 1 h. After washing with PBS, DAPI counterstaining was performed at room temperature. Apoptotic cells were imaged using a confocal microscope (Eclipse Ti2, Nikon, Tokyo, Japan).

### 2.6. Serum Biochemical Analysis

The activities of aspartate aminotransferase (AST), alanine aminotransferase (ALT), and alkaline phosphatase (ALP) and the contents of triglyceride (TG), total cholesterol (TC), high-density lipoprotein cholesterol (HDL-C), and low-density lipoprotein cholesterol (LDL-C) in serum were measured by commercial kits (Nanjing Jiancheng Bioengineering Institute, Nanjing, China).

### 2.7. Liver Antioxidant and Oxidative Biomarkers

Liver samples were homogenized in ice-cold PBS using a homogenizer (XU-YM-24, Xi Niu Co., Ltd., Shanghai, China). The resultant homogenate was centrifuged at 3000 rpm for 30 min at 4 °C. The supernatant was promptly transferred into sterile tubes and stored at −80 °C until further analysis. The activities of total antioxidant capacity (T-AOC), total superoxide dismutase (T-SOD), glutathione S-transferase (GST), glutathione peroxidase (GSH-Px), peroxidase (POD), and catalase (CAT), as well as the concentrations of malondialdehyde (MDA) and hydrogen peroxide (H_2_O_2_) were assessed using commercial kits (Nanjing Jiancheng Bioengineering Institute, Nanjing, China).

### 2.8. Quantitative Real-Time PCR

Total RNA was extracted from liver tissues employing a total RNA extraction kit (TransGen Biotech Co. Ltd, Beijing, China). The integrity and concentration of the extracted RNA were assessed through agarose gel electrophoresis and spectrophotometric analysis. Complementary DNA (cDNA) synthesis was performed utilizing a reverse transcription kit (Takara, Dalian, China). The quantification of gene expression was conducted using the SYBR Green quantitative PCR Master Mix (Takara, Dalian, China) and analyzed with the 2^−ΔΔCt^ method. β-actin served as the reference gene for the normalization of target gene expression levels. The primer sequences for the target genes are detailed in Appendix A.

### 2.9. Determination of AFB_1_ Residues and AFB_1_-DNA Adduct Levels

AFB_1_ residues in liver samples were extracted utilizing the total aflatoxin immunoaffinity column (Clover Technology Group, Beijing, China), following the manufacturer’s protocol. The extracted samples containing AFB_1_ were subsequently analyzed through high-performance liquid chromatography (HPLC). Specifically, the samples were filtered using a 0.22 μm RC filter, and a 20 μL aliquot was injected into the HPLC system. The detection of AFB_1_ was conducted using excitation and emission wavelengths of 360 nm and 440 nm, respectively. The mobile phase was a methanol–water mixture (45:55, *v*/*v*) with a flow rate of 1 mL min^−1^. The AFB_1_-DNA adduct contents in liver were determined by a commercial kit (HB253-NC, Hengyuan Biological Institute, Shanghai, China).

### 2.10. Statistical Analysis

Data were analyzed by a one-way ANOVA followed by Tukey’s multiple comparisons test using the IBM SPSS Statistics software (v26.0). Before conducting the ANOVA, the normality of the data was confirmed via the Shapiro–Wilk test and homogeneity of variances via the Levene’s test. A *p*-value of less than 0.05 was considered statistically significant. All data were expressed as mean ± standard error of mean (SEM). GraphPad Prism v7.0 was used to draw all the graphs.

## 3. Results

### 3.1. Aflatoxin Oxidase CotA Improved the Growth Performance of Japanese Quails Fed with AFB_1_-Contaminated Diet

The initial body weight of the quails selected for the experiment did not differ significantly (*p* > 0.05, Table 1). After a 21-day feeding trial, a significant reduction (*p* < 0.01) in the final body weight and ADG was observed in quails from the AFB_1_ group compared to the CON group. The addition of aflatoxin oxidase CotA to the AFB_1_-contaminated diet ameliorated the decrease in the final body weight and ADG. Moreover, there were no notable differences (*p* > 0.05) in ADFI and FCR among the different groups of quails.

### 3.2. Aflatoxin Oxidase CotA Protected Japanese Quails from AFB_1_-Induced Liver Injury

As depicted in Figure 1A, the liver histopathological sections from the CON group and the AFB_1_+CotA group displayed no notable pathological alterations, whereas the AFB_1_ group exhibited pronounced vacuolar degeneration within hepatocytes and notable inflammatory cell infiltration. The significant increase in the liver index of quails was also observed following AFB_1_ exposure (*p* < 0.05, Figure 1B). Moreover, serum ALT, AST, and ALP activities were significantly higher in the AFB_1_ group compared to the CON group (*p* < 0.05, Figure 1C–E), further indicating hepatocyte damage. Dietary supplementation with aflatoxin oxidase CotA protected quails from AFB_1_-induced hepatotoxicity, as indicated by the normalization of these key serum liver function biomarkers.

### 3.3. Aflatoxin Oxidase CotA Alleviated AFB_1_-Induced Oxidative Stress in Liver

The effect of dietary supplementation with aflatoxin oxidase CotA on the hepatic redox homeostasis of Japanese quails fed with AFB_1_-contaminated diet was investigated. As presented in Table 2, hepatic T-AOC level and the activities of antioxidant enzymes, including GST, GSH-Px, POD, and CAT, were significantly reduced in quails exposed to AFB_1_ compared to the CON group (*p* < 0.05). Concurrently, the AFB_1_ group showed a significant increase in liver H_2_O_2_ and MDA levels (*p* < 0.05). The addition of aflatoxin oxidase CotA enhanced antioxidant enzyme activities and reduced H_2_O_2_ and MDA levels in the hepatic tissues of quails compared to the AFB_1_ group. Moreover, AFB_1_ exposure significantly down-regulated Nrf2, HO-1, NQO1, and SOD1 mRNA expression (*p* < 0.05), which was mitigated by the dietary inclusion of aflatoxin oxidase CotA (Figure 2).

### 3.4. Aflatoxin Oxidase CotA Ameliorated AFB_1_-Induced Lipid Metabolism Disorder in Liver

Oil Red staining revealed extensive hepatocellular vacuoles due to lipid droplets in the AFB_1_ group (Figure 3A), aligning with the increased serum TG levels in quails consuming an AFB_1_-contaminated diet (*p* < 0.05, Figure 3B). No significant difference in serum TC content was found between the AFB_1_ group and the CON group (*p* > 0.05, Figure 3C). Moreover, quails in the AFB_1_ group exhibited higher serum LDL-C level and lower serum HDL-C level than the CON group (*p* < 0.05, Figure 3D,E). The addition of aflatoxin oxidase CotA to AFB_1_-contaminated diet effectively reversed these biochemical alterations.

### 3.5. Aflatoxin Oxidase CotA Mitigated AFB_1_-Induced Liver Apoptosis

The apoptosis rate in quail hepatocytes was detected via the TUNEL assay (Figure 4A). The AFB_1_ group exhibited a significant increase in apoptotic cells (positive-stained red) compared to the CON group. The addition of aflatoxin oxidase CotA to the AFB_1_ diet significantly reduced apoptotic cells in the liver. As shown in Figure 4B–E, compared with the CON group, the AFB_1_ group exhibited a significant increase (*p* < 0.05) in the mRNA expression levels of Bax, caspase-3, and caspase-9, while the expression of the anti-apoptotic gene Bcl-2 significantly decreased (*p* < 0.05). The addition of aflatoxin oxidase CotA significantly (*p* < 0.05) decreased the mRNA expression of Bax, caspase-3, and caspase-9 compared to the AFB_1_ group. The AFB_1_+CotA group exhibited a significant (*p* < 0.05) increase in the Bcl-2 mRNA expression level compared to the AFB_1_ group.

### 3.6. Aflatoxin Oxidase CotA Reduced AFB_1_ Residues and AFB_1_-DNA Adduct Content in Liver

The impact of dietary aflatoxin oxidase CotA on AFB_1_ residues and AFB_1_-DNA adduct content in the liver of quails is illustrated in Figure 5. In the control group, neither AFB_1_ residues nor AFB_1_-DNA adduct were detectable in the liver. In comparison to the AFB_1_ group, there was a significant reduction (*p* < 0.05) in both AFB_1_ residues and AFB_1_-DNA adduct content in the liver tissues of quails in the AFB_1_+CotA group.

## 4. Discussion

AFB_1_ contamination in feed poses a substantial toxicological threat to poultry, resulting in decreased body weight, lower feed intake, and liver damage [21]. Consequently, the development and implementation of effective AFB_1_ detoxification strategies are of paramount importance to the poultry industry. Multiple strategies, encompassing chemical, physical, and biological techniques, have been investigated to mitigate the harmful impact of AFB_1_ in poultry. Among these, biological methods have emerged as particularly promising, offering the potential to counteract AFB_1_ toxicity without inducing negative side effects in birds. Notably, CotA laccase, a novel aflatoxin oxidase, has demonstrated the capability to degrade AFB_1_ in both experimental and natural settings [12,16]. The present study was conducted to investigate whether the dietary supplementation of aflatoxin oxidase CotA could effectively ameliorate aflatoxicosis in Japanese quails. In this study, quails exposed to 50 μg kg^−1^ of AFB_1_ exhibited significant reduction in both final body weight and average daily gain. These metrics are among the most prevalent clinical manifestations observed following the ingestion of AFB_1_-contaminated diets in poultry models. Ma et al. [16] investigated ducks consuming a diet with 20 μg kg^−1^ of AFB_1_ and documented a notable decrease in average daily gain and body weight at 28 d of age. Similarly, broilers subjected to 40 μg kg^−1^ of AFB_1_ in diet for 42 d led to a notable decrease in body weight gain [22]. The inability to achieve the expected body weight can result in economic losses. The growth depression observed in quails exposed to AFB_1_ was mitigated by the dietary inclusion of aflatoxin oxidase CotA in this study. The liver, as the principal site of AFB_1_ metabolism, is also the primary target for its pathological impacts. Histological examination using H&E staining revealed increased hepatic morphological alterations and structural disarray in quails fed an AFB_1_-containing diet. Consistent with histopathological image analysis, the AFB_1_-treated group exhibited significantly higher liver index and serum liver enzyme activities (ALT, AST, and ALP) than the control group. The findings collectively demonstrate that AFB_1_ induces significant hepatic damage, thereby validating the successful establishment of the quail AFB_1_ poisoning model. The incorporation of aflatoxin oxidase CotA into the AFB_1_-contaminated diet mitigates hepatic damage, normalizes tissue morphology, and restores serum liver enzyme activity. Thus, dietary supplementation with CotA exhibits a substantial protective effect against AFB_1_-induced hepatic injuries.

Oxidative stress, characterized by an imbalance between oxidants and antioxidants within the body, is a pivotal factor in the pathogenesis of AFB_1_-induced hepatic damage [21]. T-AOC serves as a marker reflecting the antioxidant status, while antioxidant enzymes, including SOD, GST, GSH-Px, POD, and CAT, are crucial for eliminating free radicals and peroxides [23]. H_2_O_2_ is formed starting from the superoxide anion (O_2_^−^), and MDA is the end product of lipid peroxidation, both of which are commonly employed as indicators of oxidative stress [24,25]. In the present study, quails exposed to AFB_1_ showed signs of oxidative stress, indicated by a notable rise in hepatic H_2_O_2_ and MDA levels, alongside a marked decrease in antioxidant enzymes, including T-AOC, GST, GSH-Px, POD, and CAT. These findings align with prior research showing that AFB_1_ exposure significantly elevated reactive oxygen species (ROS), H_2_O_2_, and MDA levels while suppressing T-AOC, SOD, and CAT activities in the livers of mice, thereby suggesting that oxidative stress may serve as the initial trigger in exacerbating AFB_1_-induced liver injury [26]. The supplementation of aflatoxin oxidase CotA in the diet of quails enhanced T-AOC, GST, GSH-Px, POD, and CAT activities and lowered H_2_O_2_ and MDA concentrations in the liver, thereby mitigating oxidative stress induced by AFB_1_. Nrf2 is a key transcription factor involved in oxidative stress caused by AFB_1_ [27,28]. Intracellular oxidative stress prompts the translocation of Nrf2 to the nucleus, where it attaches to the antioxidant response element to control the expression of antioxidant genes, such as HO-1, NQO1, and SOD [29]. The present study demonstrated that exposure to AFB_1_ inhibited the mRNA expression of Nrf2 and its downstream genes HO-1, NQO1, and SOD in the liver of quails. These findings align with earlier research showing that AFB_1_ exposure inhibits Nrf2 nuclear translocation [27,28]. Dietary supplementation with aflatoxin oxidase CotA ameliorated the AFB_1_-induced suppression of Nrf2 and enhanced antioxidant gene expression.

Changes in lipids and lipoproteins contribute to the development of aflatoxicosis [30]. In this study, AFB_1_ exposure was found to cause hepatic steatosis in quails, marked by excessive lipid accumulation in liver tissues. Moreover, AFB_1_ exposure elevated serum TG and LDL-C levels and reduced HDL-C level. The mechanisms by which AFB_1_ induces lipid metabolism disorders are multifaceted. A previous study has revealed that AFB_1_ can increase cyclooxygenase-2 (COX-2) expression, subsequently raising mitophagy levels and disrupting normal mitochondrial lipid metabolism [31]. Additionally, AFB_1_ exposure has been linked to changes in bile acid metabolism, which is crucial for lipid digestion and absorption [32]. In addition to these direct effects on lipid metabolism, AFB_1_ also interacts with other metabolic pathways. For example, AFB_1_ exposure in Hep3B cells has been shown to induce dynamic metabolic reprogramming, affecting pathways such as fatty acid synthesis and oxidation, glycerophospholipid metabolism, and the tricarboxylic acid (TCA) cycle [33]. These alterations further contribute to the overall disruption of lipid metabolism in the liver. The dietary supplementation of aflatoxin oxidase CotA could ameliorate the negative impact of AFB_1_ on lipid metabolism.

Apoptosis, an essential and programmed form of cell death, is integral to the normal physiological processes in avian species. Prior studies have shown that AFB_1_ can compromise the structural integrity of hepatocyte mitochondria, reduce mitochondrial membrane potential, and initiate mitochondrion-dependent apoptotic pathways [26,28]. In alignment with these findings, TUNEL staining revealed that AFB_1_ treatment markedly elevated hepatocyte apoptosis compared to the CON group in this study. Conversely, the AFB_1_+CotA group exhibited a notable decrease in the liver cell apoptosis index relative to the AFB_1_ group. Mitochondrial apoptosis is modulated by the Bcl-2 protein family, which includes the pro-apoptotic Bax and anti-apoptotic Bcl-2 [34]. The anti-apoptotic protein Bcl-2 inhibits apoptosis by neutralizing Bax, which facilitates the permeabilization of the mitochondrial outer membrane, a vital step in the intrinsic apoptosis pathway. Upon release from mitochondria, the pro-apoptotic molecule cytochrome c (Cyt-C) interacts with apoptotic protease activating factor 1 (Apaf-1) to form apoptosome complex. This complex subsequently activates pro-caspase-9, which in turn converts pro-caspase-3 into caspase-3, culminating in the execution phase of apoptosis. Prior research has demonstrated that AFB_1_ can enhance the mRNA expression of Bax, caspase-9, and caspase-3, while reducing Bcl-2 mRNA expression level, thereby promoting cellular apoptosis [26,35]. This study further assessed the expression profiles of mitochondrial pathway-related apoptosis genes in the liver tissue of quails. The findings reveal the significant up-regulation of Bax, caspase-9, and caspase-3 mRNA expression, alongside a notable down-regulation of Bcl-2 mRNA level in the AFB_1_ group in comparison with the CON group. Moreover, the dietary supplementation of aflatoxin oxidase CotA in the AFB_1_ diet notably decreased the mRNA expression levels of Bax, caspase-9, and caspase-3, while enhancing Bcl-2 mRNA expression. These results indicate that aflatoxin oxidase CotA effectively mitigated AFB_1_-induced hepatocyte apoptosis in quails.

The liver is the primary site for AFB_1_ metabolism, where cytochrome P450 enzymes convert AFB_1_ into exo-AFB_1_-8,9-epoxide (AFBO) [4]. Subsequently, AFBO interacts with biomacromolecules such as DNA, resulting in AFB_1_-DNA adduct formation [36]. These adducts serve as promising biomarkers for assessing AFB_1_ exposure and AFBO production in animals [5]. Our study found that dietary supplementation with aflatoxin oxidase CotA significantly reduced AFB_1_ residues and AFB_1_-DNA adducts in the liver of quails exposed to AFB_1_. These findings collectively suggest that aflatoxin oxidase CotA possesses a robust AFB_1_ degradation capability within the gastrointestinal tract of quails, thereby reducing the concentration of AFB_1_ absorbed by enterocytes and maintaining normal hepatic function.

## 5. Conclusions

In summary, Japanese quails fed with diet contaminated with 50 μg kg^−1^ of AFB_1_ for 21 days presented reduced growth performance. The hepatoxicity of AFB_1_ may closely associate with oxidative stress, lipid metabolism disorder, and apoptosis. The dietary supplementation of aflatoxin oxidase CotA could effectively detoxify aflatoxicosis in Japanese quails fed with AFB_1_-contaminated diet and reduced AFB_1_ residues and AFB_1_-DNA adduct content in the liver. These results emphasize the potential application of aflatoxin oxidase CotA as a promising AFB_1_ detoxification agent in the feed industry. Given that the feeding trial in this study was limited to a duration of three weeks, it is imperative to investigate further the long-term protective effects of aflatoxin oxidase CotA in Japanese quails exposed to AFB_1_. Additionally, further research is warranted to assess the AFB_1_-detoxifying efficacy of CotA in other poultry species.

## Figures and Tables

**Figure 1 animals-15-01555-f001:**
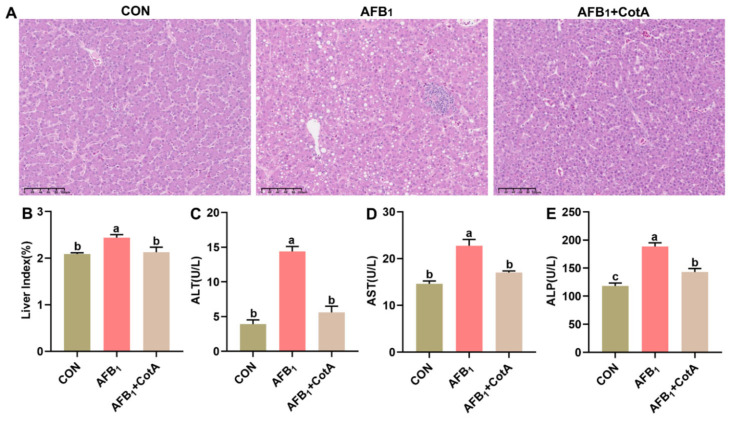
Protective effect of aflatoxin oxidase CotA on AFB_1_-induced liver injury of Japanese quails. (**A**) The liver histopathology observed using HE staining. (**B**) Liver index. (**C**) Serum alanine aminotransferase (ALT) activity. (**D**) Serum aspartate aminotransferase (AST) activity. (**E**) Serum alkaline phosphatase (ALP) activity. Values are expressed as mean ± SEM, and different lowercase letters indicate a significant difference at *p* < 0.05. CON = fed with basal diet; AFB_1_ = fed with basal diet + 50 μg kg^−1^ of AFB_1_; AFB_1_+CotA = fed with basal diet + 50 μg kg^−1^ of AFB_1_ + 1U kg^−1^ of aflatoxin oxidase CotA.

**Figure 2 animals-15-01555-f002:**
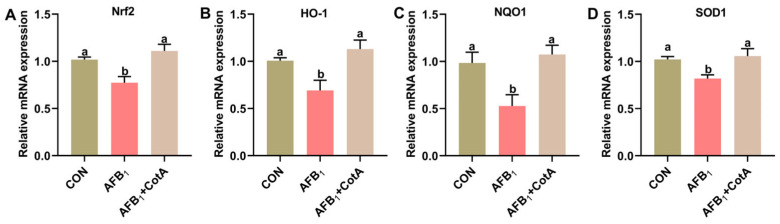
The mRNA expression of the transcription factor Nrf2 and its downstream genes in the liver of quails. (**A**–**D**) The relative expression of Nrf2, HO-1, NQO1, and SOD1. Values are expressed as mean ± SEM, and different lowercase letters indicate a significant difference at *p* < 0.05. CON = fed with basal diet; AFB_1_ = fed with basal diet + 50 μg kg^−1^ of AFB_1_; AFB_1_+CotA = fed with basal diet + 50 μg kg^−1^ of AFB_1_ + 1U kg^−1^ of aflatoxin oxidase CotA.

**Figure 3 animals-15-01555-f003:**
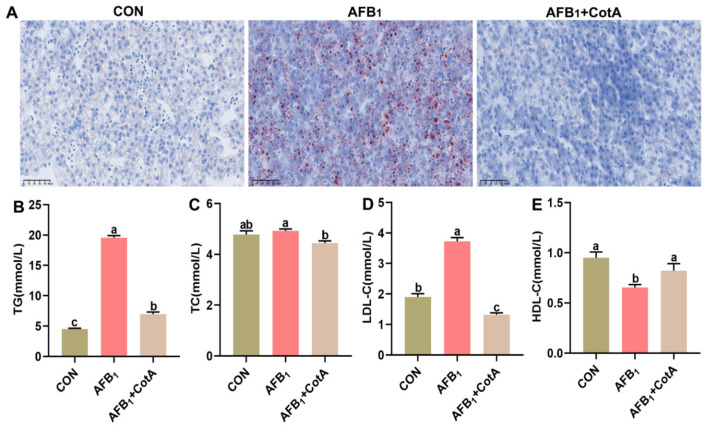
Effect of aflatoxin oxidase CotA on the hepatic lipid accumulation in Japanese quails fed with AFB_1_-contaminated diet. (**A**) Oil Red O staining of liver tissue sections. (**B**) Serum triglyceride (TG) content. (**C**) Serum total cholesterol (TC) content. (**D**) Serum low-density lipoprotein cholesterol (LDL-C) content. (**E**) Serum high-density lipoprotein cholesterol (HDL-C) content. Values are expressed as mean ± SEM, and different lowercase letters indicate a significant difference at *p* < 0.05. CON = fed with basal diet; AFB_1_ = fed with basal diet + 50 μg kg^−1^ of AFB_1_; AFB_1_+CotA = fed with basal diet + 50 μg kg^−1^ of AFB_1_ + 1U kg^−1^ of aflatoxin oxidase CotA.

**Figure 4 animals-15-01555-f004:**
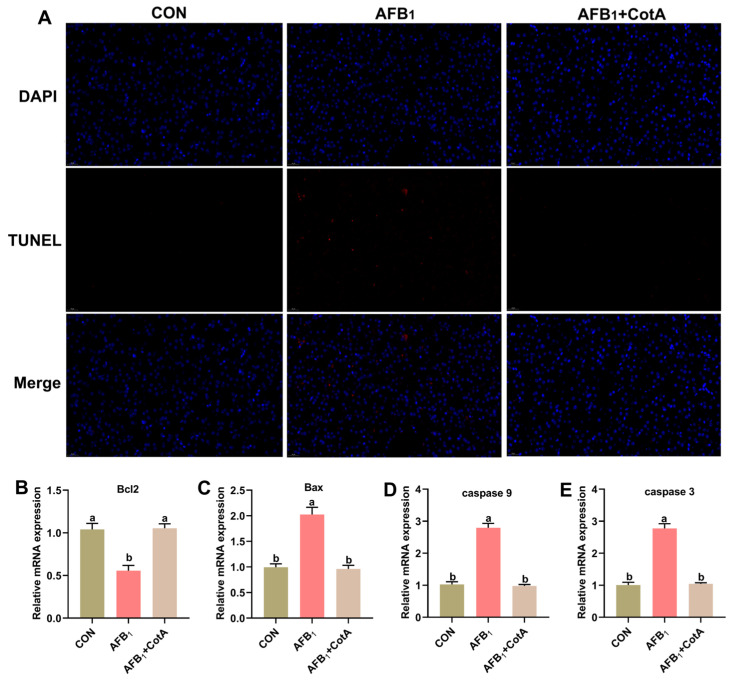
Effect of aflatoxin oxidase CotA on hepatocyte apoptosis and apoptosis-related gene expression of Japanese quails fed with AFB_1_-contaminated diet. (**A**) TUNEL stained paraffin sections from liver tissues. The relative mRNA expression of (**B**) B-cell lymphoma 2 (Bcl-2), (**C**) Bcl-2-associated X protein (Bax), (**D**) caspase 9, and (**E**) caspase 3 in the liver. Values are expressed as mean ± SEM, and different lowercase letters indicate a significant difference at *p* < 0.05. CON = fed with basal diet; AFB_1_ = fed with basal diet + 50 μg kg^−1^ of AFB_1_; AFB_1_+CotA = fed with basal diet + 50 μg kg^−1^ of AFB_1_ + 1U kg^−1^ of aflatoxin oxidase CotA.

**Figure 5 animals-15-01555-f005:**
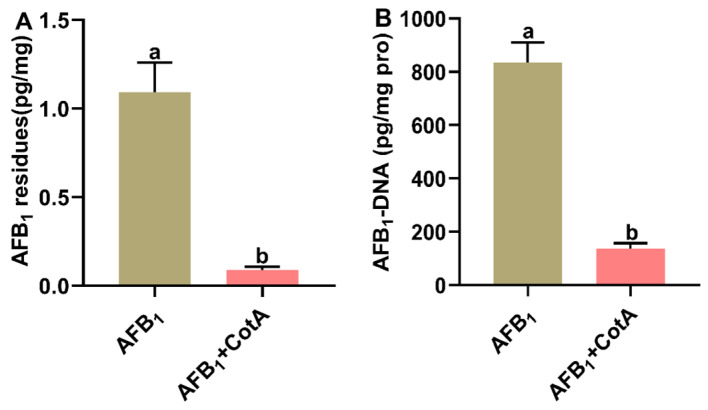
Aflatoxin oxidase CotA reduced AFB_1_ residues and AFB_1_-DNA adduct content in liver of quails. (**A**) AFB_1_ residues in liver. (**B**) AFB_1_-DNA adduct content in liver. Values are expressed as mean ± SEM, and different lowercase letters indicate a significant difference at *p* < 0.05. CON = fed with basal diet; AFB_1_ = fed with basal diet + 50 μg kg^−1^ of AFB_1_; AFB_1_+CotA = fed with basal diet + 50 μg kg^−1^ of AFB_1_ + 1U kg^−1^ of aflatoxin oxidase CotA.

**Table 1 animals-15-01555-t001:** Effect of dietary supplementation with aflatoxin oxidase CotA on the growth performance of Japanese quails fed with AFB_1_-contaminated diet.

Items ^1^	Treatment ^2^			SEM	*p*-Value
	CON	AFB_1_	AFB_1_+CotA		
Initial BW (g)	69.43	69.52	69.84	0.10	0.188
Final BW (g)	156.63 ^a^	149.40 ^b^	158.57 ^a^	1.43	0.009
ADG (g)	4.15 ^a^	3.80 ^b^	4.23 ^a^	0.07	0.009
ADFI (g)	17.38	17.09	18.25	0.27	0.203
FCR	4.19	4.51	4.32	0.09	0.381

^1^ BW: bodyweight; ADG: average daily gain; ADFI: average daily feed intake; FCR: feed conversion ratio. ^2^ CON = fed with basal diet; AFB_1_ = fed with basal diet + 50 μg kg^−1^ of AFB_1_; AFB_1_+CotA = fed with basal diet + 50 μg kg^−1^ of AFB_1_ + 1U kg^−1^ of aflatoxin oxidase CotA. ^a, b^ Means in the same row with different superscripts are significantly different (*p* < 0.05).

**Table 2 animals-15-01555-t002:** Effect of dietary supplementation with aflatoxin oxidase CotA on antioxidant and oxidative stress-related biomarkers in liver tissues of Japanese quails fed with AFB_1_-contaminated diet.

Item ^1^	Treatment ^2^			SEM	*p*-Value
	CON	AFB_1_	AFB_1_+CotA		
T-AOC (mmol/g prot)	1.71 ^a^	1.26 ^b^	1.94 ^a^	0.10	0.004
T-SOD (U/mg prot)	17.99	14.58	18.43	0.97	0.216
GST (U/mg prot)	117.77 ^a^	78.45 ^b^	91.80 ^ab^	5.78	0.006
GSH-Px (U/mg prot)	95.50 ^a^	84.04 ^b^	90.77 ^a^	1.48	0.001
POD (U/mg prot)	7.48 ^a^	4.54 ^b^	8.39 ^a^	0.55	0.002
CAT (U/mg prot)	14.55 ^a^	10.05 ^b^	15.40 ^a^	0.88	0.014
H_2_O_2_ (mmol/g prot)	3.77 ^a^	8.81 ^b^	4.73 ^a^	0.74	0.003
MDA (nmol/mg prot)	12.63 ^b^	18.58 ^a^	12.43 ^b^	1.06	0.012

^1^ T-AOC: total antioxidant capacity; T-SOD: total superoxide dismutase; GST: glutathione S-transferase; GSH-Px: glutathione peroxidase; POD: peroxidase; CAT: catalase; H_2_O_2_: hydrogen peroxide; MDA: malondialdehyde. ^2^ CON = fed with basal diet; AFB_1_ = fed with basal diet + 50 μg kg^−1^ of AFB_1_; AFB_1_+CotA = fed with basal diet + 50 μg kg^−1^ of AFB_1_ + 1U kg^−1^ of aflatoxin oxidase CotA. ^a, b^ Means in the same row with different superscripts are significantly different (*p* < 0.05).

## Data Availability

All data in this study are available from the authors upon reasonable request.

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
