# Peer review of "Dietary Supplementation of Novel Aflatoxin Oxidase CotA Alleviates Aflatoxin B_1_-Induced Oxidative Stress, Lipid Metabolism Disorder, and Apoptosis in the Liver of Japanese Quails"

_animals, 2025, doi:10.3390/ani15111555_

Round 1

Reviewer 1 Report

Comments and Suggestions for Authors

I have read the manuscript entitled “Dietary supplementation of novel aflatoxin oxidase CotA alleviates aflatoxin B1-induced oxidative stress, lipid metabolism disorder and apoptosis in liver of Japanese quails,” submitted for review to the journal Animals, in order to assess its suitability for publication. The authors have chosen an important and timely topic, addressing a novel strategy for the detoxification of aflatoxin B1 using the CotA enzyme in Japanese quails. The subject fits very well within the scope of Animals. The selected methods are appropriate, and the results and discussion are well described. However, before the article can be accepted for publication, it requires significant improvement in English language usage, as well as minor additions to the Methods section and the Introduction.

Detailed comments:

1). The introduction is well written and provides a solid background on the issue of aflatoxin B1 in poultry production. My only remark is the lack of references to recent studies, especially those focusing on in vivo detoxification in different species. It would also be beneficial to emphasize the uniqueness of the present study in the context of existing literature.

2). The methodology used in the study is standard and described in a clear and detailed manner. However, the manuscript would benefit from the inclusion of information regarding the number of animals used and the basis for their selection. Additionally, more details on the CotA enzyme—such as specific activity, biochemical characteristics, stability, and purity—should be provided. The statistical methods applied by the authors are appropriate, but it would be useful to mention how the assumptions of ANOVA were verified.

3). The results are very well presented, and the tables and figures are clear, with appropriate indications of significance levels. The only suggestion here is to avoid repeating numerical data from the tables directly in the main text.

4). The discussion is generally well written but somewhat uncritical. Some literature descriptions could be shortened and better integrated with the study results. Moreover, the authors should moderate their conclusions related to less significant findings and clearly acknowledge the limitations of the study, such as its duration and potential species-specific effects.

5). The conclusions presented in the manuscript are appropriate to the results and summarize that CotA has potential as a feed additive supporting AFB1 detoxification. However, these conclusions should be expressed with more caution. It would also be valuable to highlight the need for further studies, for example, involving other poultry species.

6). Regarding the references, I did not detect any signs of plagiarism. Although there are instances of self-citation, they appear justified. The cited literature is relevant and appropriate.

7). Unfortunately, the English language throughout the manuscript requires substantial revision. There are numerous grammatical and syntactical errors, as well as imprecise expressions. I strongly recommend having the manuscript thoroughly edited by a native speaker.

Therefore, my recommendation is: Minor Revision.

Author Response

Comments 1: The introduction is well written and provides a solid background on the issue of aflatoxin B1 in poultry production. My only remark is the lack of references to recent studies, especially those focusing on in vivo detoxification in different species. It would also be beneficial to emphasize the uniqueness of the present study in the context of existing literature.

Response 1: Thanks for your comment. We added a more recent reference about AFB1 detoxification in the revised manuscript.

15 Liu, M.; Zhao, L.; Gong, G.; Zhang, L.; Shi, L.; Dai, J.; Han, Y.; Wu, Y.; Khalil, M.M.; Sun, L. Invited review: Remediation strategies for mycotoxin control in feed. J. Anim. Sci. Biotechnol. 2022, 13, 19.

Comments 2: The methodology used in the study is standard and described in a clear and detailed manner. However, the manuscript would benefit from the inclusion of information regarding the number of animals used and the basis for their selection. Additionally, more details on the CotA enzyme—such as specific activity, biochemical characteristics, stability, and purity—should be provided. The statistical methods applied by the authors are appropriate, but it would be useful to mention how the assumptions of ANOVA were verified.

Response 2: Thanks for your comments.

Japanese quail chicks (1-week of age, female) were purchased from Henan Aoxiang Quail Breeding Base (Henan, China). After acclimation for 1 week, a total of 225 Japanese quail chicks were randomly allocated into three groups, each comprising five replicates with 15 quail chicks per replicate.

The enzymatic properties of aflatoxin oxidase CotA were documented in our previous with good pH stability, thermostability and high catalytic efficiency to AFB1 as can be seen in reference 12 and 20 in this manuscript.

Before using ANOVA, normality of the data was confirmed via Shapiro-Wilk test and homogeneity of variances via Levene's test.

Comments 3: The results are very well presented, and the tables and figures are clear, with appropriate indications of significance levels. The only suggestion here is to avoid repeating numerical data from the tables directly in the main text.

Response 3: Thanks for your comment. The numerical data from the tables were not repeated in the revised manuscript.

Comments 4: The discussion is generally well written but somewhat uncritical. Some literature descriptions could be shortened and better integrated with the study results. Moreover, the authors should moderate their conclusions related to less significant findings and clearly acknowledge the limitations of the study, such as its duration and potential species-specific effects.

Response 4: Thanks for your comment. The limitation of this study was added in the conclusion in the revised manuscript “Given that the feeding trial in this study was limited to a duration of three weeks, it is imperative to further investigate the long-term protective effects of aflatoxin oxidase CotA in Japanese quails exposed to AFB1.”

Comments 5: The conclusions presented in the manuscript are appropriate to the results and summarize that CotA has potential as a feed additive supporting AFB1 detoxification. However, these conclusions should be expressed with more caution. It would also be valuable to highlight the need for further studies, for example, involving other poultry species.

Response 5: Thanks for your comment. We added in the conclusion that further research is warranted to assess the AFB1-detoxifying efficacy of CotA in other poultry species.

Comments 6: Regarding the references, I did not detect any signs of plagiarism. Although there are instances of self-citation, they appear justified. The cited literature is relevant and appropriate.

Response 6: Thanks for your comment.

Comments 7: Unfortunately, the English language throughout the manuscript requires substantial revision. There are numerous grammatical and syntactical errors, as well as imprecise expressions. I strongly recommend having the manuscript thoroughly edited by a native speaker.

Response 7: Thanks for your comment. We have asked for help from a native speaker to correct grammatical and syntactical errors and improve the English writing of the current manuscript.

Reviewer 2 Report

Comments and Suggestions for Authors

Keywords line 43: typo in the keyword section: “Live damage” should be “Liver damage”.

Introduction line 47-49. No reference provided to the claim that aflatoxin led to the deaths of more than 100,000 turkeys. Also, the statement “Aflatoxins were initially identified in the early 1960s” seems broad. The particular year it was first discovered/identified should be mentioned and by whom. I believe this will be a better opening for the introduction.

Line 59-60 mentions that there is an urgent necessity to explore effective strategies for preventing AFB1 deleterious impacts on poultry. Not that it is compulsory but a reference like this or something else that is similar will help strengthen the sentence: https://doi.org/10.1016/j.toxicon.2023.107262

Section 2.1. The reasoning behind inclusion levels in treatment group 2 and 3 were not mentioned. Is this the pilot study or is the inclusion level based on previous findings in the lab. If so, a reference will be appropriate if such previous findings have been published.

Also, section 2.1 mentioned using all female chicks. What is the rationale behind this? Does the aflatoxin stress not affect the male birds?

Additionally, why was the 21-day feeding trial used, that is 3 weeks for birds that were 2 weeks old before the start of the experiment. The birds are 5 weeks at experiment termination. If it is based on preliminary data or previous lab publications, it should be noted in section 2.1. or any other appropriate section.

  1. Is there a reason why a longer schedule was not used and why 3 weeks?
  2. Why not day of hatch birds and the experiment used 2 weeks.
  3. The paper does not mention is the quails were hatched onsite or procured from a hatchery. If procured, was there any adjustment to the new environment before the start of trials.

Lack of Positive Control. Including a known commercial mycotoxin binder or antioxidant would have strengthened the conclusions by providing a benchmark for CotA’s effectiveness. Since it wasn’t included and it is probably too late for that, it should be noted for future experiments and maybe included in the discussion why it was not included in this experiment.

While P-values and SEM are presented, a more detailed explanation of how replicates were handled statistically would strengthen section 2.10.

Author Response

Comments 1: Keywords line 43: type in the keyword section: “Live damage” should be “Liver damage”.

Response 1: Thanks for your comment. We have made change.

Comments 2: Introduction line 47-49. No reference provided to the claim that aflatoxin led to the deaths of more than 100,000 turkeys. Also, the statement “Aflatoxins were initially identified in the early 1960s” seems broad. The particular year it was first discovered/identified should be mentioned and by whom. I believe this will be a better opening for the introduction.

Response 2: Thanks for your comment. The references about the cause of the death of more than 100,000 turkeys and the discover of aflatoxins by Van der Zijden et al. in 1961 were added in the revised manuscript.

  1. Van der Zijden, A.S.M.; Koelensmid, W.; Boldingh, J.; Barrett, C.B.; Ord, W.O.; Philip, J. Aspergillus flavus and Turkey X disease: Isolation in crystalline form of a toxin responsible for Turkey X-disease. Nature 1962, 195, 1060-
  2. Blount, W.P. Turkey “X” disease. Br. Turkey 1961, 9, 55-58.

Comments 3: Line 59-60 mentions that there is an urgent necessity to explore effective strategies for preventing AFB1 deleterious impacts on poultry. Not that it is compulsory but a reference like this or something else that is similar will help strengthen the sentence: https://doi.org/10.1016/j.toxicon.2023.107262

Response 3: We agreed with the review. The reference mentioned has been added in the revised manuscript to strengthen the statement that there is an urgent necessity to explore effective strategies for preventing or alleviating the deleterious impacts of AFB1 on poultry.

7 Wang, Y.; Wang, X.; Li, Q. Aflatoxin B1 in poultry liver: Toxic mechanism. Toxicon 2023, 233, 107262.

Comments 4: Section 2.1. The reasoning behind inclusion levels in treatment group 2 and 3 were not mentioned. Is this the pilot study or is the inclusion level based on previous findings in the lab. If so, a reference will be appropriate if such previous findings have been published.

Response 4: Thanks for your comment. The AFB1 level in diet and the dose of aflatoxin oxidase CotA included in this study was set according to our previous study investigating the protective effect of dietary CotA in ducks fed with AFB1 contaminated diet [16]. We have added reference in this section.

Comments 5: Also, section 2.1 mentioned using all female chicks. What is the rationale behind this? Does the aflatoxin stress not affect the male birds?

Response 5: Thanks for your comment. Indeed, no special attention was paid on the gender of quail chicks in this study. Both male and female birds were vulnerable to the toxic impacts of AFB1.

Comments 6: Additionally, why was the 21-day feeding trial used, that is 3 weeks for birds that were 2 weeks old before the start of the experiment. The birds are 5 weeks at experiment termination. If it is based on preliminary data or previous lab publications, it should be noted in section 2.1. or any other appropriate section.

  1. Is there a reason why a longer schedule was not used and why 3 weeks?
  2. Why not day of hatch birds and the experiment used 2 weeks.
  3. The paper does not mention is the quails were hatched onsite or procured from a hatchery. If procured, was there any adjustment to the new environment before the start of trials.

Response 6: Thanks for your comments.

The feeding experiment started at 2-week of age and ended at 5-week of age. Obvious liver injury including vacuolar degeneration within hepatocytes, extensive hepatocellular vacuoles and significant increase in apoptosis were observed after feeding Japanese quails with AFB1-contaminated diet for 21 days. Thus, the duration of the 21-day feeding trial was enough the evaluate the short-term protective effect of dietary aflatoxin oxidase CotA. Moreover, we added in the conclusion “Given that the feeding trial in this study was limited to a duration of three weeks, it is imperative to further investigate the long-term protective effects of aflatoxin oxidase CotA in Japanese quails exposed to AFB1.”

Japanese quail chicks (1-week of age, female) were purchased from Henan Aoxiang Quail Breeding Base (Henan, China). After acclimation for 1 week, a total of 225 Japanese quail chicks were randomly allocated into three groups, each comprising five replicates with 15 quail chicks per replicate.

Comments 7: Lack of Positive Control. Including a known commercial mycotoxin binder or antioxidant would have strengthened the conclusions by providing a benchmark for CotA’s effectiveness. Since it wasn’t included and it is probably too late for that, it should be noted for future experiments and maybe included in the discussion why it was not included in this experiment.

Response 7: Thanks for your comment. We agreed with the review that including a known commercial mycotoxin binder or antioxidant would have strengthened the conclusions by providing a benchmark for CotA’s effectiveness. We added in the conclusion that further research is warranted to assess the AFB1-detoxifying efficacy of CotA in other poultry species.

Comments 8: While P-values and SEM are presented, a more detailed explanation of how replicates were handled statistically would strengthen section 2.10.

Response 8: Thanks for your comment. Before using ANOVA, normality of the data was confirmed via Shapiro-Wilk test and homogeneity of variances via Levene's test.